# Analyzing the Time Spectrum of Supernova Neutrinos to Constrain Their Effective Mass or Lorentz Invariance Violation

**Celio A. Moura** *,†, **Lucas Quintino** † and **Fernando Rossi-Torres** †

Centro de Ciências Naturais e Humanas, Universidade Federal do ABC—UFABC,
Santo André 09210-580, SP, Brazil; lucas.quintino@ufabc.edu.br (L.Q.); f.torres@ufabc.edu.br (F.R.-T.)
* Correspondence: celio.moura@ufabc.edu.br; Tel.: +55-11-4996-7960
† These authors contributed equally to this work.

**Abstract:** We analyze the expected arrival time spectrum of supernova neutrinos using simulated luminosity and compute the expected number of events in future detectors such as the DUNE Far Detector and Hyper-Kamiokande. We develop a general method using minimum square statistics that can compute the sensitivity to any variable affecting neutrino time of flight. We apply this method in two different situations: First, we compare the time spectrum changes due to different neutrino mass values to put limits on electron (anti)neutrino effective mass. Second, we constrain Lorentz invariance violation through the mass scale, $M_{QG}$, at which it would occur. We consider two main neutrino detection techniques: 1. DUNE-like liquid argon TPC, for which the main detection channel is $\nu_e + {}^{40}\mathrm{Ar} \to e^- + {}^{40}\mathrm{K}^*$, related to the supernova neutronization burst; and 2. HyperK-like water Cherenkov detector, for which $\bar{\nu}_e + p \to e^+ + n$ is the main detection channel. We consider a fixed supernova distance of 10 kpc and two different masses of the progenitor star: (i) 15 $M_\odot$ with neutrino emission time up to 0.3 s and (ii) 11.2 $M_\odot$ with neutrino emission time up to 10 s. The best mass limits at $3\sigma$ are for $\mathcal{O}(1)$ eV. For $\nu_e$, the best limit comes from a DUNE-like detector if the mass ordering happens to be inverted. For $\bar{\nu}_e$, the best limit comes from a HyperK-like detector. The best limit for the Lorentz invariance violation mass scale at the $3\sigma$ level considering a superluminal or subluminal effect is $M_{QG} \gtrsim 10^{13}$ GeV ($M_{QG} \gtrsim 5 \times 10^5$ GeV) for linear (quadratic) energy dependence.

**Keywords:** neutrino; supernova; Lorentz invariance violation; mass

## 1. Introduction

One of the most important and still open questions in neutrino physics is the absolute value of neutrino masses. In addition to this question, we do not have a completely proven model of a mass generation mechanism that could explain the existence of this non-zero mass. Despite the lack of a complete picture of the neutrino mass generation mechanism, in recent years, we have seen great progress in neutrino physics, especially coming from oscillation experiments [1]. These experiments point to the fact that neutrino mass eigenstates mix during propagation, giving rise to neutrino flavor oscillation. The present oscillation experimental data cannot point out which mass eigenstate is the lightest one, with two options being possible: normal ordering (NO), $m_1 < m_2 < m_3$; or inverted ordering (IO), $m_3 < m_1 < m_2$. Oscillation experiments usually measure $|\Delta m_{ij}^2|$, the mass squared differences in the module; and $\theta_{ji}$, the mixing angles of the mass eigenstates, where $i, j = 1, 2, 3$ and $i > j$. From the matter effect in oscillation, we do know that $\Delta m_{21}^2$ is positive. Recent values of the oscillation parameters, with $1\sigma$ uncertainty, can be found in, e.g., Ref. [1].

One of the possible sources of neutrinos is the collapse of stars. This process emits around 99% of the star's internal energy in the form of neutrinos and antineutrinos with average energy on the order of 10 MeV. Detection of the supernova (SN) neutrino burst is crucial in order to understand the stellar explosion mechanism, and it can provide an early warning for electromagnetic observation experiments [2].

During the stellar collapse that leads to the explosion of an SN, neutrino emission occurs in different stages. The first stage is neutronization, in which a fusion reaction between electrons and protons produces neutrons and electron neutrinos ($\nu_e$). This process produces a peak in the $\nu_e$ luminosity curve as a function of time that lasts for about 25 ms. During neutronization, $\nu_e$s are trapped behind the shock wave formed by the collapse and are only released when the matter density becomes sufficiently low. The second stage is accretion, in which the matter from the collapsing star is attracted to the newly formed neutron star, producing neutrinos of all flavors. In this stage, all types of neutrinos are emitted. The third stage is cooling, which occurs after the SN explosion. During cooling, the neutron star releases all types of neutrinos and decreases its binding energy. For reviews of SN neutrinos and neutrino emission properties, see Refs. [3,4]. The impact of neutrino mass ordering—NO or IO—is relevant for the measurement of the neutronization burst [5].

There are several ways to measure the neutrino mass, and important constraints already exist. Before we discuss our estimate using SN neutrinos, we list a few constraints from different techniques.

From the high-energy spectrum range of tritium beta decay, the Katrin experiment found a limit of 0.8 eV at 90% C.L. [6]. Other nuclei, such as $^{187}$Re and $^{163}$Ho, may also $\beta$ decay and provide limits [7]. From the electron capture decays of $^{163}$Ho ($^{163}$Ho$^+ + e^- \rightarrow ^{163}$Dy $+ \nu_e$), $m_{\nu_e} < 225$ eV at 95% C.L. [8] and $m_{\nu_e} < 460$ eV at 68% C.L. [9]. In the MANU experiment, with $^{187}$Re $\beta$ decay, an upper limit of 26 eV at 95% C.L. was found [10]. The MiBeta Collaboration obtained an upper limit of 15 eV at 90% C.L. [11]. Recently, the Project 8 Collaboration, using the Cyclotron Radiation Emission Spectroscopy (CRES) technique and a cm$^3$-scale physical detection volume, obtained $m_\nu < 152$ eV from the continuous tritium beta spectrum [12].

Despite the uncertainties in the nuclear matrix elements from neutrinoless double $\beta$ decay ($(Z, A) \rightarrow (Z + 2, A) + e^- + e^-$) [13], experiments that study $0\nu\beta\beta$ from $^{136}$Xe nuclei, such as the KamLAND-ZEN [14] and EXO-200 [15], constrained, respectively, $m_{ee} < 0.061 - 0.165$ eV and $m_{ee} < 0.093 - 0.286$ eV. In this lepton-number violating process, the decay rate of the nuclei scales with the effective neutrino mass, where the neutrino is a Majorana fermion. The Heidelberg–Moscow collaboration claimed evidence of this phenomenon with $m_{ee} \approx 0.3$ eV, $m_{ee}$ representing the coherent sum of masses ($m_i$) of the mass eigenstates [16].

Cosmology also imposes limits on neutrino masses. Neutrinos have large free-streaming lengths that depend on their masses. They smear out fluctuations that are imprinted in the cosmic microwave background and in galaxies. From the cosmic microwave background, using the Planck satellite, $m_\nu < 0.09$ eV at 95% C.L. [17], even though there are several degeneracies related to the cosmological model parameters [18].

All bounds described above are summarized in Table 1.

**Table 1.** Neutrino mass limits for different experiments and different experimental methods. Except where it is written otherwise, the confidence intervals are given at 90% C.L.

| Experiment | Method | Mass Limit |
|---|---|---|
| Katrin [6] | Tritium beta decay | 0.8 eV |
| Springer et al. [8][1] | $e^-$ capture decays of $^{163}$Ho | 225 eV at 95% C.L |
| MANU [10] | $^{187}$Re $\beta$ decay | 26 eV at 95% C.L. |
| MiBeta [11] | $^{187}$Re $\beta$ decay | 15 eV |
| Project 8 [12] | CRES | 152 eV |
| KamLAND-ZEN [14] | $0\nu\beta\beta$ | 0.061–0.165 eV |
| EXO-200 [15] | $0\nu\beta\beta$ | 0.093–0.286 eV |
| Planck [17] | Cosmic microwave | 0.09 eV at 95% C.L. |

Neutrino mass bounds can be obtained from type II SN bursts, which have short duration and happen at astronomical distances. The idea is that neutrinos travel long

distances in a time that depends on their mass as well as on their energy. So there is a delay compared to the time of flight of a supposedly massless particle that impacts the time spectrum of the neutrino events at the Earth [19,20].

Using SN1987A inverse $\beta$ decay data ($\bar{\nu}_e + p \rightarrow n + e^+$) from Kamiokande [21], IMB [22], and Baksan [23], an upper limit of $m_\nu < 5.8$ eV at 95% C.L. [24] was obtained using a proper likelihood of event-by-event analysis [25]. Lamb and Loredo, in a similar analysis, constrained the neutrino mass to be less than 5.7 eV [26]. The future Jiangmen Underground Neutrino Observatory (JUNO) [27], a 20 kton liquid-scintillator detector using the inverse beta decay channel, may limit the neutrino mass with an SN at 10 kpc distance and NO as $m_\nu < (0.83 \pm 0.24)$ eV at 95% C.L. A sub-eV mass limit for a future SN may be reached in several detectors using a time-structured signal, as demonstrated in Ref. [28]. Ref. [29] set limits of the order of a few tenths of eV at 95% C.L. for an SN distance of 10 kpc, considering a different stellar explosion scenario where a QCD phase transition induces the SN explosion [30].

A Liquid Argon Time Projection Chamber (LArTPC), such as the far detector of the future Deep Underground Neutrino Experiment (DUNE) [31,32], is sensitive to the detection of electron neutrinos ($\nu_e$) from SNs [33]. Depending on the type of neutrino mass ordering, such a detector can be important to prove the existence of the neutronization process in explosions of massive stars. A detector with 40 kton of liquid argon can detect thousands of electron neutrino events with energies around 10 MeV through charged current (CC) interactions. Furthermore, the short temporal duration of the neutronization peak may place more restrictive constraints on physical parameters associated with arrival-time spectrum changes. It is expected to have a sensitivity of around 0.90-2.00 eV for the neutrino mass [33,34].

Cherenkov light detectors, sensitive to $\bar{\nu}_e$, can probe the entire neutrino emission period of an SN event, making it possible to establish limits on neutrino mass [24,26]. Hyper-Kamiokande (HyperK) [35]—the next generation and improved Super-Kamiokande (SK)—will work with a water Cherenkov detector (WtCh) and is very sensitive to the detection of $\bar{\nu}_e$ via $\bar{\nu}_e + p \rightarrow n + e^+$. For HyperK, it is expected to have an absolute mass sensitivity from 0.5 to 1.3 eV [35].

In this work, we obtain limits for $\nu_e$ and $\bar{\nu}_e$ effective masses again, and we compare the sensitivities of an LArTPC and a WtCh.

Using the same methodology applied for the effect of the neutrino mass, one can analyze the time spectrum of an SN event and impose constraints on Lorentz invariance violation (LIV). Since SN neutrinos can travel long distances, they can be good probes of possible quantum gravity fluctuations in space–time. These fluctuations may generate energy-dependent modification in the neutrino velocity [36,37]. For a review on the effects of LIV in astrophysical neutrinos, such as SN neutrinos, and for a more-recent phenomenological discussion, see Refs. [38,39].

Inspired by [40], we conduct an analysis to constrain a quantum gravity mass scale parameter. In this analysis, we separate the two effects of mass and LIV because the electron (anti)neutrino effective mass may still be relatively small compared to the upper limits that we find, which means that the effect of neutrino mass may be small enough so that we can study the LIV effect without considering the neutrino mass. We explore the time spectrum changes caused, independently, by neutrino mass or LIV along the neutrino's propagation from the SN to the Earth. We consider two different masses of progenitor stars at a distance of 10 kpc and two types of detectors: a DUNE-like 40 kton LArTPC and a HyperK-like 100 kton WtCh detector. The two mass orderings are considered in order to explore the different sensitivities to neutrino mass and LIV energy scale.

This article is organized as follows: In Section 2, we show how to evaluate the neutrino fluxes in SNs and their respective fluxes at Earth after oscillations. In Section 3, we discuss the method developed to evaluate the number of events and the calculation of the squared function, $\chi^2$, minimized in order to put bounds on the neutrino mass or LIV. Section 4 presents some details of the experiments we simulate in our study and their related numbers

of events per channel of detection. In Section 5, we present our main results for the limits of neutrino mass or LIV and discuss them. Finally, in Section 6, we present our main conclusions and future perspectives.

## 2. Neutrino Fluxes

Inside a star, each neutrino flavor $\nu_\beta$ ($\beta = e, \mu, \tau$) has a differential flux at time $t$ after the bounce of the SN core, described as [3]

$$\frac{d^2\Phi^0_{\nu_\beta}}{dtdE} = \frac{L_{\nu_\beta}(t)}{4\pi d^2} \frac{f_{\nu_\beta}(t, E)}{\langle E_{\nu_\beta}(t)\rangle},$$

(1)

where $\Phi^0_{\nu_\beta}$ are the original fluxes for each $\nu_\beta$, $L_{\nu_\beta}(t)$ are the neutrino flavor luminosities, $d$ is the SN distance to the Earth detector, $f_{\nu_\beta}(t, E)$ are the neutrino energy spectra, $\langle E_{\nu_\beta}(t)\rangle$ are the mean neutrino energies, and $E$ is the instant neutrino energy. $L_{\nu_\beta}(t)$, $\langle E_{\nu_\beta}(t)\rangle$, and $f_{\nu_\beta}(t, E)$ depend on the SN model.

We consider two models with different progenitor masses: one with 15 solar masses (15 $M_\odot$) [41] and one with 11.2 $M_\odot$ [42]. Figure 1a,b show the $\nu_e$ (black solid line), $\bar{\nu}_e$ (red dotted line), and $\nu_x$ (blue dashed line) luminosity time evolution for the two models; $\nu_x$ is defined as the sum of all muon and tau neutrinos and antineutrinos, i.e., $\nu_\mu + \bar{\nu}_\mu + \nu_\tau + \bar{\nu}_\tau$.

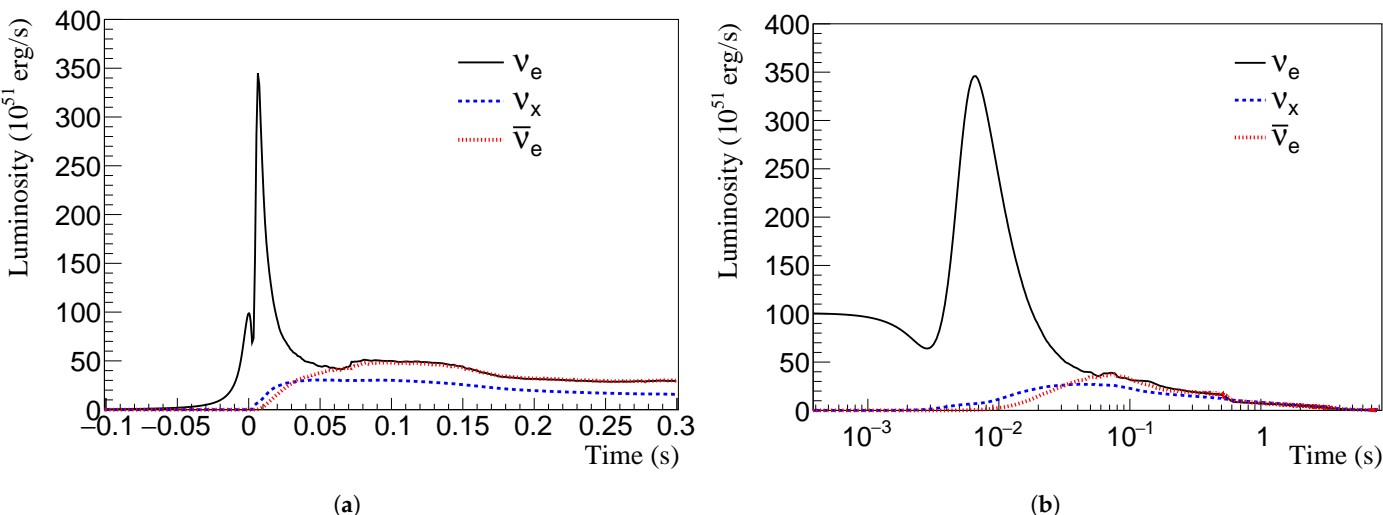

(**a**)  (**b**)

**Figure 1.** Luminosity time evolution for the two SN progenitor star models analyzed in this work. Black solid line—$\nu_e$, red dotted line—$\bar{\nu}_e$, and blue dashed line—$\nu_x = \nu_\mu + \bar{\nu}_\mu + \nu_\tau + \bar{\nu}_\tau$. (**a**) 15 $M_\odot$ SN [41]. (**b**) 11.2 $M_\odot$ SN [42].

The neutrino energy spectra can be parameterized as [43]

$$f_{\nu_\beta}(t, E) = \lambda_\beta(t) \left(\frac{E}{\langle E_{\nu_\beta}(t)\rangle}\right)^{\alpha_\beta(t)} \exp\left(-\frac{[\alpha_\beta(t) + 1]E}{\langle E_{\nu_\beta}(t)\rangle}\right),$$

(2)

where $\alpha_\beta(t)$ is the so-called pinching parameter that is model-dependent and accounts for the variations in the quasi-thermal spectrum; $\lambda_\beta(t)$ is the time-dependent normalization factor, so that $\int_E f_{\nu_\beta}(t, E) \, dE = 1$.

Inside the star environment, neutrinos interact with electrons, protons, and neutrons, suffering flavor conversion in the resonances (low and high) according to the MSW effect [44–46]. After crossing the stellar matter, neutrinos travel incoherently through vacuum until they are detected at Earth. The expressions for the oscillated differential fluxes, which are shown below, are obtained from Ref. [47].

The differential $\nu_e$ flux at Earth for NO is

$$\frac{d^2\Phi_{\nu_e}}{dtdE} = \frac{d^2\Phi_{\nu_x}^0}{dtdE} , \tag{3}$$

while for the flavors $\nu_\mu$, $\nu_\tau$, $\bar{\nu}_\mu$, and $\bar{\nu}_\tau$, represented by $\nu_x$, it can be written as

$$4\frac{d^2\Phi_{\nu_x}}{dtdE} = \frac{d^2\Phi_{\nu_e}^0}{dtdE} + \sin^2\theta_{12}\frac{d^2\Phi_{\bar{\nu}_e}^0}{dtdE} + (2 + \cos^2\theta_{12})\frac{d^2\Phi_{\nu_x}^0}{dtdE} , \tag{4}$$

where $\theta_{12}$ is the mixing angle between $\nu_1$ and $\nu_2$ mass eigenstates.

The differential $\bar{\nu}_e$ flux is

$$\frac{d^2\Phi_{\bar{\nu}_e}}{dtdE} = \cos^2\theta_{12}\frac{d^2\Phi_{\bar{\nu}_e}^0}{dtdE} + \sin^2\theta_{12}\frac{d^2\Phi_{\nu_x}^0}{dtdE} . \tag{5}$$

The mixing angle $\theta_{13}$ is considered small compared to $\theta_{12}$, and the expressions above are evaluated accordingly. The $\theta_{12}$ value is: $33.45^{\circ+0.77^\circ}_{\phantom{\circ}-0.75^\circ}$ for NO and $33.45^{\circ+0.78^\circ}_{\phantom{\circ}-0.75^\circ}$ for IO [1].

Now, for IO, we can write the following differential fluxes for $\nu_e$, $\nu_x$, and $\bar{\nu}_e$, respectively:

$$\frac{d^2\Phi_{\nu_e}}{dtdE} = \sin^2\theta_{12}\frac{d^2\Phi_{\nu_e}^0}{dtdE} + \cos^2\theta_{12}\frac{d^2\Phi_{\nu_x}^0}{dtdE} , \tag{6}$$

$$4\frac{d^2\Phi_{\nu_x}}{dtdE} = \cos^2\theta_{12}\frac{d^2\Phi_{\nu_e}^0}{dtdE} + \frac{d^2\Phi_{\bar{\nu}_e}^0}{dtdE} + (2 + \sin^2\theta_{12})\frac{d^2\Phi_{\nu_x}^0}{dtdE} , \tag{7}$$

and

$$\frac{d^2\Phi_{\bar{\nu}_e}}{dtdE} = \frac{d^2\Phi_{\bar{\nu}_x}^0}{dtdE} . \tag{8}$$

We do not consider nonlinear collective effects in this work. They are not particularly relevant in the neutronization burst [3] and would bring unnecessary complication to the oscillation treatment. In Ref. [48], the authors considered Earth matter effects on neutrino oscillation and their possible effects on the neutrino masses; however, such effects do not considerably affect their results—see their Table I. For the current status of neutrino flavor conversion in high-density environments and its relevance in astrophysical systems such as SNs, see Ref. [49].

## 3. Methodology

In this section, we describe our basic methodology in order to compute the number of events using the neutrino fluxes produced in SNs introduced in the previous Section 2. We explain how to compute the number of events when one includes the modification to time propagation caused by mass or LIV. After the calculation of the number of events, we present the minimum square statistical analysis used to constrain the neutrino mass or LIV.

### 3.1. Number of Events

The event rate, $R(t, E)$, the number of neutrino events in a given detector per energy and time units, is evaluated as

$$R(t, E) = n_t\sigma(E)\epsilon(E)\frac{d^2\Phi_{\nu_\beta}}{dtdE} , \tag{9}$$

where $n_t$ is the number of targets in the detector, and $\sigma(E)$ and $\epsilon(E)$ are, respectively, the cross section and the detector efficiency, both of which depend on the neutrino energy. The actual computation, which includes the detector material, efficiency, and each process interaction cross-section is conducted by the fast event-rate calculation tool SNOwGLoBES [50].

The standard time of detection for a massless particle is given by

$$t_{det} = t_{em} + d/c \,, \tag{10}$$

where $t_{em}$ is the time of emission and $d/c$ is the travel time from the source to the detector. The time change due to mass or a nonstandard effect such as LIV is approximately

$$t'_{det} = t_{em} + d/c + \Delta t(E) \,, \tag{11}$$

where $\Delta t$ depends on the neutrino energy and is calculated by the mass effect or by a nonstandard effect such as LIV. So the arrival time difference in the neutrino flux is

$$t'_{det} - t_{det} = \Delta t(E) \,. \tag{12}$$

We integrate Equation (9) in energy bins of 0.2 MeV in the energy interval from 0 to 100 MeV, obtaining the number of events per time unit in each energy bin

$$\left(\frac{dN}{dt}\right)_{E_i,t} = \int_{E_i-\delta E}^{E_i+\delta E} R(t,E)dE \,, \tag{13}$$

where $E_i$ is the average energy of the *i*-th energy bin, $E_1 = 0.1$ MeV, and $\delta E = 0.1$ MeV is half of the energy interval of the bin. This gives the energy spectrum for standard physics and massless neutrinos. In order to obtain the time spectrum, i.e., the number of events per time interval, we compute

$$N_j = \int_{t_j}^{t_j+\delta t} \sum_i \left(\frac{dN}{dt}\right)_{E_i,t} dt \,, \tag{14}$$

where $N_j$ and $t_j$ are the event number and the initial time of the *j*-th time bin, respectively, and $\delta t$ is the bin width. Considering the effect of propagation time delay,

$$\left(\frac{dN}{dt'}\right)_{E_i,t'} = \left(\frac{dN}{dt}\right)_{E_i,t+\Delta t_i} = \int_{E_i-\delta E}^{E_i+\delta E} R(t',E)dE \,, \tag{15}$$

The time delay, $\Delta t_i$, for each energy bin, considering its average, $E_i$, is given by Equation (12). The effect of the different delay for each bin of the energy spectrum is to spread neutrinos that are emitted in a given time through different time bins. The number of events in a given time bin is

$$N'_j = \int_{t_j}^{t_j+\delta t} \sum_i \left(\frac{dN}{dt}\right)_{E_i,t+\Delta t_i} dt \,, \tag{16}$$

where we sum over all the events per time unit for which the delay brings them to the *j*-th time bin.

### 3.2. Time Delay

We consider the time delay during neutrino propagation in two different situations:

First, neutrinos from galactic and extra-galactic sources experience a modification in their times of flight because of their masses [19,20]. The delay caused by the neutrino mass, $m_\nu$, can be written as

$$\Delta t = \frac{d}{2}\left(\frac{m_\nu}{E}\right)^2 \,, \tag{17}$$

where $E$ is the neutrino energy and $d$ is the distance from the neutrino source to the detector. Figure 2a shows the delay, in seconds, as a function of energy, considering different values of neutrino masses and $d = 10$ kpc. The black solid line is for $m_\nu = 1.0$ eV and the blue dash-dotted line is for $m_\nu = 2.0$ eV. This plot clearly demonstrates the effect of the neutrino

mass and neutrino energy on the time of flight for a given distance: the combination of large mass and low energy generates longer delays.

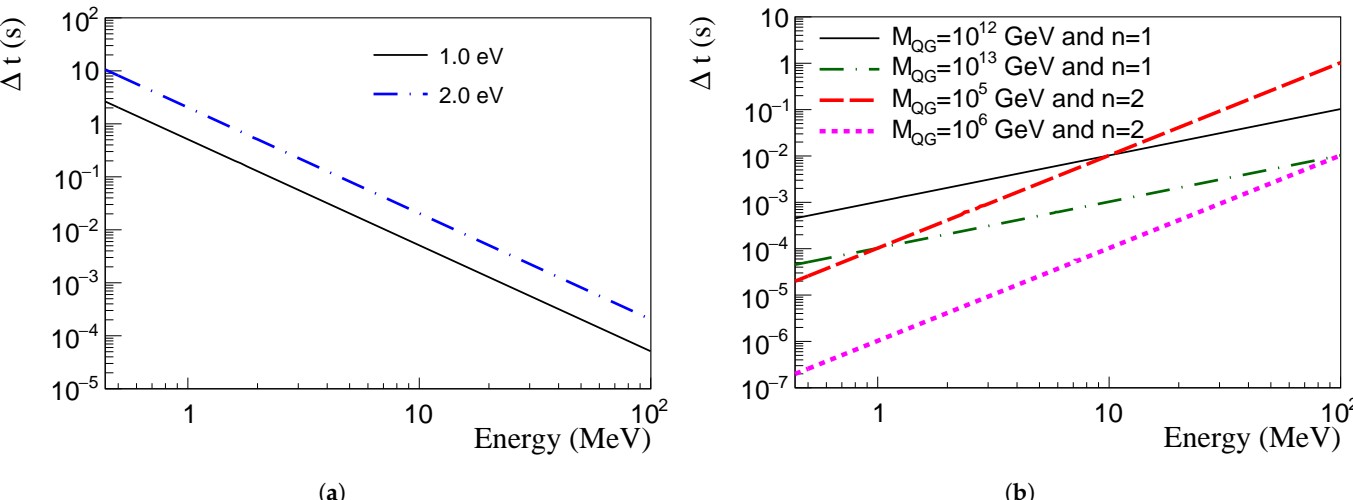

**Figure 2.** Delay, $\Delta t$, in seconds, considering a source at distance $d = 10$ kpc for different neutrino masses (left plot) and different LIV energy scales (right plot) as a function of neutrino energy. Neutrino mass values are presented in the left plot. LIV energy scale and energy dependence, $n$, are presented in the right plot. See also the text for details. (**a**) Neutrino mass effect. (**b**) LIV effect for different energy scales.

Second, LIV may cause time modification along neutrino propagation. Here, we consider LIV without violation of $CPT$ symmetry. In this kind of model, the dispersion relation of particles is affected during their propagation by the interaction with the medium [36,37]. When we translate this in a temporal effect, there is a time change that can be written as [40]

$$\Delta t = \pm d \left( \frac{E}{M_{QG}} \right)^n , \tag{18}$$

where $M_{QG}$ is the energy (mass) scale that corresponds to the Lorentz symmetry breaking of a high-energy theory; $n$ is the order of the effect, which we consider linear or quadratic in energy; and the $+$ or $-$ signs correspond to subluminal (delay) or superluminal (advance) differences in the expected detection time, respectively.

Figure 2b shows $\Delta t$ for different LIV energy dependences and scales: linear dependence ($n = 1$)—$M_{QG} = 10^{12}$ GeV (solid black line) and $M_{QG} = 10^{13}$ GeV (green dash-dotted line); quadratic dependence ($n = 2$)—$M_{QG} = 10^5$ GeV (dashed red line) and $M_{QG} = 10^6$ GeV (dotted magenta line). The effect of LIV is opposite to the effect of mass: $\Delta t$ is larger when $E$ increases. As expected, a higher energy scale, $M_{QG}$, decreases $\Delta t$, making the sensitivity to LIV lower.

Limits for the parameters considered in this work—mass ($m_\nu$) and LIV scale ($M_{QG}$)—are obtained through a minimum square statistical analysis of the function

$$\chi^2 = \sum_i \frac{\left( N_i' - N_i \right)^2}{\sigma_i^2} , \tag{19}$$

where $N_i$ and $N_i'$ are, respectively, the numbers of events given by Equations (14) and (16), and $\sigma_i$ accounts for the uncertainties. In our analysis, we include the statistical uncertainty $\sigma_i = \sqrt{N_i}$. The confidence levels are given by $\Delta \chi^2 = \chi^2 - \chi^2_{\min} = 1$ ($1\sigma = 68.27\%$ C.L.), $\Delta \chi^2 = 4$ ($2\sigma = 95.45\%$ C.L.), and $\Delta \chi^2 = 9$ ($3\sigma = 99.73\%$ C.L.).

## 4. Detection Techniques

Before showing our results, we present some general features of the two detector types that we used in the analysis and point out a few suppositions about their role in the precision of our results.

### 4.1. Liquid Argon TPC

The main detection channel in a LArTPC is the weak CC interaction between $\nu_e$ and Ar: $\nu_e + {}^{40}\text{Ar} \rightarrow e^- + {}^{40}\text{K}^*$ [51]. The $e^-$ produced in this reaction is observable together with deexcitation products from the excited state of ${}^{40}\text{K}^*$. This reaction has a detection threshold energy of approximately 5 MeV. Other detection channels include $\bar{\nu}_e$ CC and elastic scattering (ES) on $e^-$. Neutral current (NC) scattering, $\nu + \text{Ar} \rightarrow \nu + \text{Ar}^*$, is a considerable channel, and it can be detected in the deexcitation gammas (9.8 MeV decay line) from the excited state of $\text{Ar}^*$ [33]. In our analysis, we consider the possibility of detection of all channels. We do not include any background events, uncertainties on neutrino production or propagation, or any systematics. This approach does not significantly change the sensitivity of the parameters [48].

### 4.2. Water Cherenkov Detector

The main detection channel of a water Cherenkov (WtCh) detector is the inverse $\beta$ decay: $\bar{\nu}_e + p \rightarrow e^+ + n$. Events related to this channel have a nearly isotropic distribution. The future main example of this kind of detector is HyperK, which can detect SN neutrinos with energies as low as 3 MeV and has directional sensitivity [35]. In this work, we consider a 100 kton of WtCh detector. Despite the inverse $\beta$ decay being the main channel, HyperK can detect neutrinos through other channels as well: $\nu$-e scattering ($\nu + e^- \rightarrow \nu + e^-$) and $\nu_e$ or $\bar{\nu}_e$ CC interaction with oxygen ($\nu_e + {}^{16}\text{O} \rightarrow e^- + {}^{16}\text{F}^{(*)}$ or $\bar{\nu}_e + {}^{16}\text{O} \rightarrow e^+ + {}^{16}\text{N}^{(*)}$). The $\nu$-e scattering events are strongly peaked in the direction coming from the SN. In our work, we consider that the SN events point directly to the detector. The same suppositions taken into account for the LArTPC are also considered in the WtCh case, so we can conduct an analysis under equal conditions and following the same hypotheses.

## 5. Results and Discussion

In this section, we show the limits on the neutrino masses and LIV scale, $M_{QG}$, for subluminal and superluminal cases for NO and IO for both detector types described in Section 4. The SN distance is fixed at 10 kpc, and we consider two distinct models with progenitor masses: 15 $\text{M}_\odot$ and 11.2 $\text{M}_\odot$.

In the analysis of the neutrino mass limit, we assume the use of all $\nu_e$ interaction channels for the DUNE-like detector and all $\bar{\nu}_e$ interaction channels for the HyperK-like detector. For $M_{QG}$, we sum all neutrino flavor events from all interaction channels, since LIV may equally affect the neutrino eigenstates during their propagation.

We discuss the results for the 15 $M_\odot$ SN model [41] first.

In Figure 3a,b, one sees the number of events, $N$, in a 40 kton LArTPC as a function of bins of time. Each curve shows a different detection channel. On the left (right) side of Figure 3, Figure 3a,b considers NO (IO). The $\nu_e$ interactions are represented by: (i) black solid curves—CC interaction with Argon; (ii) red dotted—NC interaction with Ar; (iii) yellow dash-dotted—ES on $e^-$. The $\bar{\nu}_e$ interactions are represented by: (i) blue dashed curves—CC interaction with Argon; (ii) green dash-dotted—NC interaction with Ar; (iii) brown dashed—ES on $e^-$. Other flavors interact with Argon by NC (violet dashed-dot curves) and ES (magenta dash-double dotted curves).

We show equivalent curves in Figure 4a,b for a 100 kton WtCh detector. The main channel in this case is the inverse beta decay (IBD) used to probe $\bar{\nu}_e$. Colored lines show $\bar{\nu}_e$ NC interaction with ${}^{16}\text{O}$ (red dotted) and ES on $e^-$ (brown dashed); $\nu_e$ interactions with ${}^{16}\text{O}$ via CC (violet dash-dotted) and NC (blue dashed), and $\nu_e$ ES on $e^-$ (yellow dash-two dotted). Other flavors interact via ES on $e^-$ (magenta dash-two dotted) and NC interaction with ${}^{16}\text{O}$ (green dash-dotted).

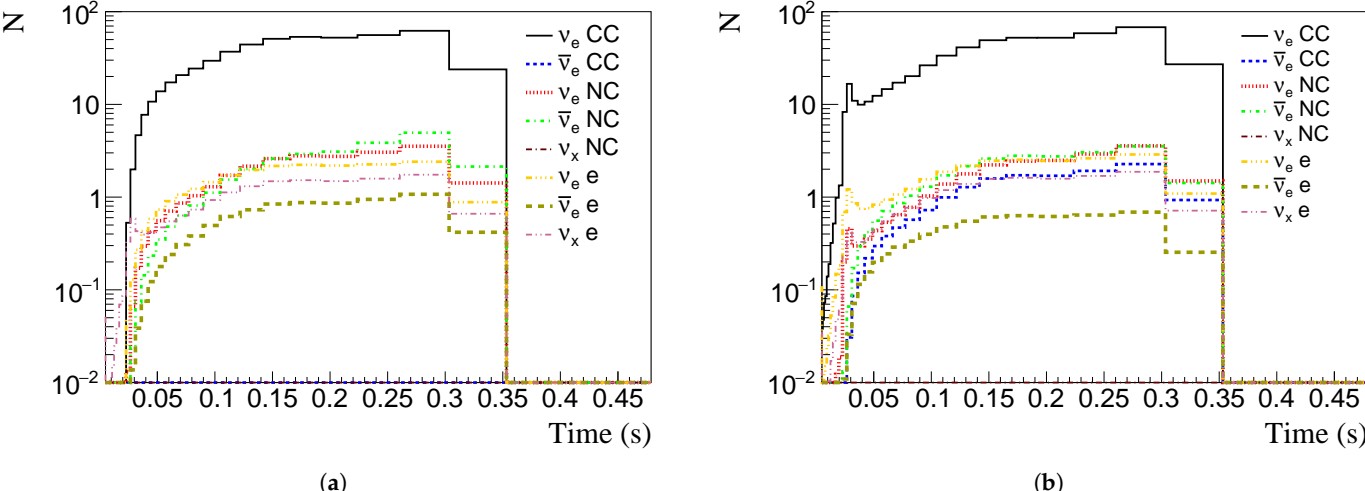

**Figure 3.** Number of events per time bin in a 40 kton LArTPC. The $\nu_e$ interactions are represented by (i) black solid—CC interaction with Argon; (ii) red dotted—NC interaction with Ar; (iii) yellow dash-dotted—ES on $e^-$. The $\bar{\nu}_e$ interactions are: (i) blue dashed curves—CC interaction with Argon; (ii) green dash-dotted—NC interaction with Ar; (iii) brown dashed—ES on $e^-$. Other flavors interact with Argon by NC (violet dashed-dot curves) and ES (magenta dash-double dotted curves). (**a**) Normal ordering. (**b**) Inverted ordering.

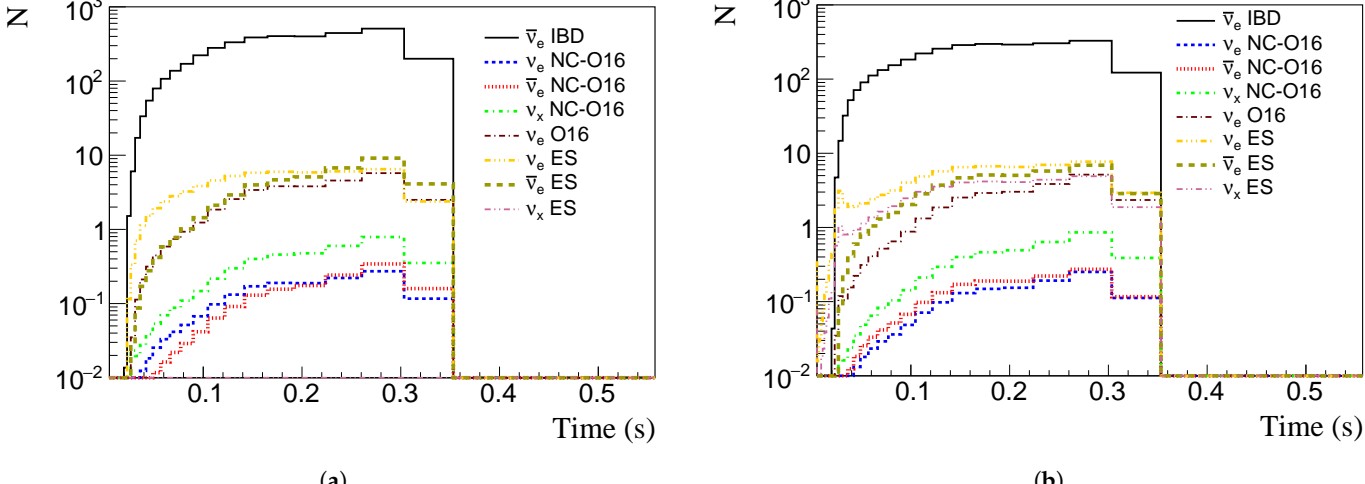

**Figure 4.** Number of events per bins of time for a 100 kton WtCh detector. Several detection channels can be distinguished: (i) for $\bar{\nu}_e$—IBD (black solid lines), NC interaction with $^{16}$O (red dotted), and ES on $e^-$ (brown dashed); (ii) $\nu_e$—CC (violet dash-dotted) and NC (blue dashed) interaction with $^{16}$O, and ES on $e^-$ (yellow dash-two dotted); (iii) $\nu_x$—ES on $e^-$ (magenta dash-two dotted) and NC interaction with $^{16}$O (green dash-dotted). (**a**) Normal ordering. (**b**) Inverted ordering.

Figure 5 shows the time distribution of the $\nu_e$ (Figure 5a,b) and $\bar{\nu}_e$ (Figure 5c,d) event numbers for a 40 kton LArTPC detector and 100 kton WtCh detector, respectively. Left (Right) plots consider NO (IO). The solid black line refers to a massless neutrino. To illustrate the time delay caused by different mass values, blue dashed refers to 1 eV neutrino mass, red dotted to 2 eV, and green dash-dotted to 4 eV.

The mass values were chosen in order to illustrate the effect of the mass in the signal time delay as described by Equation (17). Figure 5a,b show the relevance of mass ordering to probe the neutronization burst, which is suppressed in NO. On the other hand, for $\bar{\nu}_e$, Figure 5c,d, the neutronization burst is not present. We notice a larger number of events in the HyperK-like detector since it is larger in size and because the inverse $\beta$

decay cross-section, which is the main channel of interaction, is larger than the CC $\nu_e$-Ar cross-section.

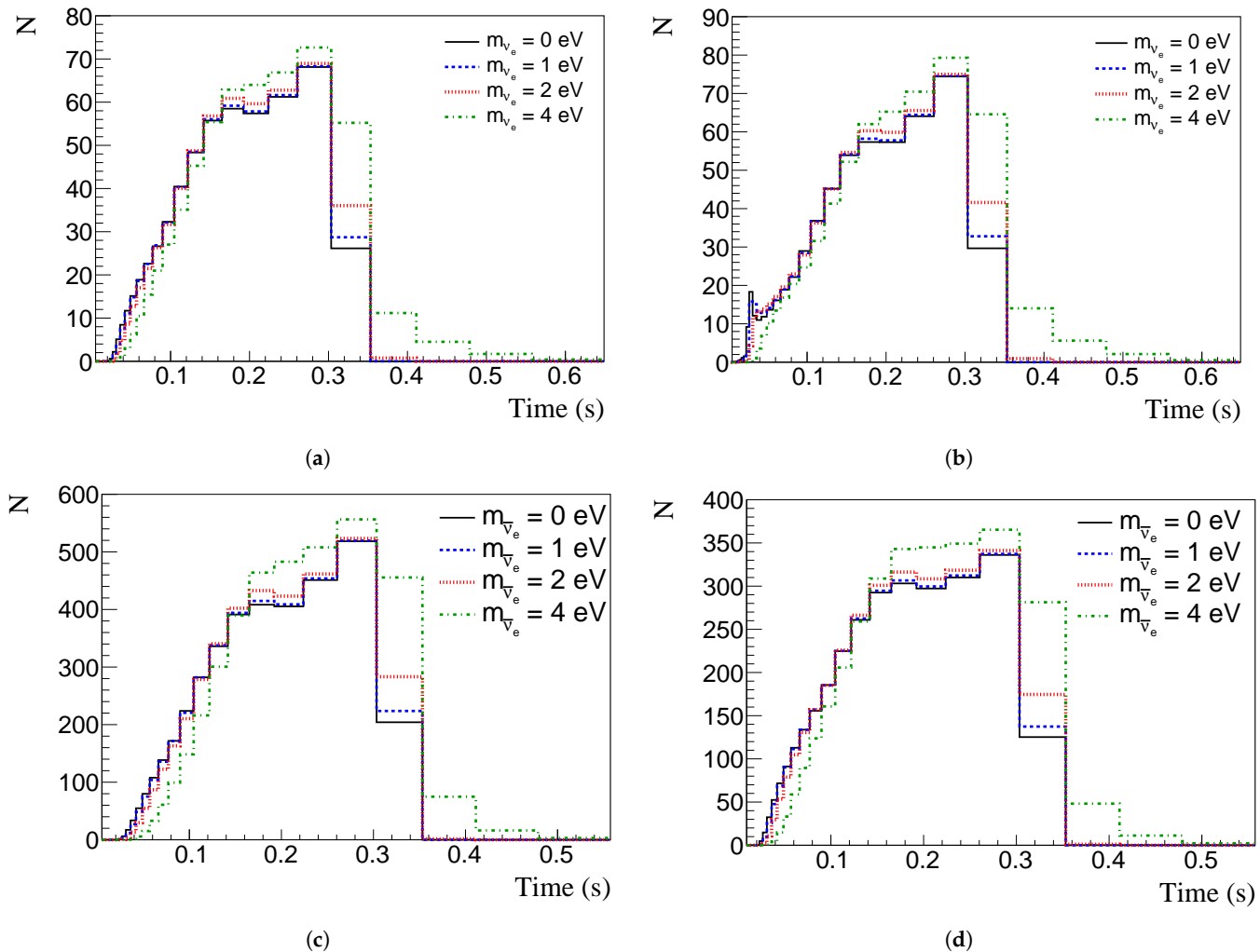

**Figure 5.** The top (bottom) panels show the number of $\nu_e$ ($\bar{\nu}_e$) events per time bin in a 40 kton LArTPC (100 kton WtCh detector). Left (right) panels consider NO (IO). Black solid line refers to massless neutrino events, blue dashed to 1 eV neutrino mass, red dotted to 2 eV, and green dash-dotted to 4 eV. (**a**) NO events in a LArTPC. (**b**) IO events in a LArTPC. (**c**) NO events in a WtCh detector. (**d**) IO events in a WtCh detector.

Figures 6–9 show the time spectrum considering NO (IO) for a 40 kton LArTPC detector and a 100 kton WtCh detector, respectively. In all figures top (bottom) panels show the time spectrum change due to subluminal (superluminal) LIV, left (right) panels consider energy dependence $n = 1$ ($n = 2$), black solid lines refer to conserved Lorentz invariance, blue dashed line to $M_{QG} = 10^{12}$ GeV ($M_{QG} = 2 \times 10^5$ GeV), red dotted line to $M_{QG} = 10^{13}$ GeV ($M_{QG} = 6 \times 10^5$ GeV), and green dash-dotted to $M_{QG} = 10^{14}$ GeV ($M_{QG} = 10^6$ GeV). We notice that the lower the value of $M_{QG}$ is, the greater the effect of LIV in $\Delta t$—see Equation (18). As we mentioned before, LIV models predict the possibility of superluminal propagation, causing an advance in the detection time. This can be seen in the bottom of Figures 6–9. A time delay similar to the effect of neutrino mass but with different energy dependence can be seen by the subluminal LIV effect in the top of Figures 6–9.

Using Equation (19), we obtain the $\Delta\chi^2$ in terms of neutrino mass and $M_{QG}$.

The mass limits are shown in Figure 10a for the LArTPC and WtCh detectors. The mass bounds for $\nu_e$ ($\bar{\nu}_e$) in a 40 kton LArTPC (100 kton WtCh) are represented by a solid black (dotted blue) line and dashed red (dash-dotted green) line, respectively, for IO and NO.

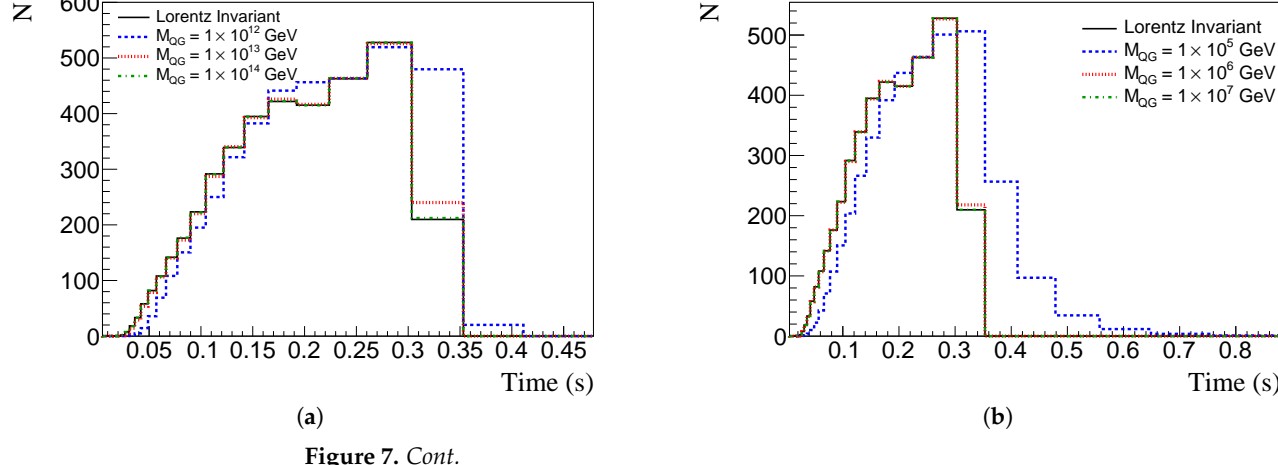

**Figure 6.** Event number per time bin in a 40 kton LArTPC considering NO, different LIV energy dependence, $n$, and different energy scale, $M_{QG}$. Top (bottom) panels show subluminal (superluminal) effect. Left (Right) panels show $n = 1$ ($n = 2$). Blue dashed lines represent event numbers for $M_{QG} = 10^{12}$ GeV ($2 \times 10^5$ GeV), red dotted for $M_{QG} = 10^{13}$ GeV ($6 \times 10^5$ GeV), and green dash-dotted for $M_{QG} = 10^{14}$ GeV ($10^6$ GeV). Black solid line refers to conserved Lorentz invariance. (**a**) Subluminal; $n = 1$. (**b**) Subluminal; $n = 2$. (**c**) Superluminal; $n = 1$. (**d**) Superluminal; $n = 2$.

**Figure 7.** *Cont.*

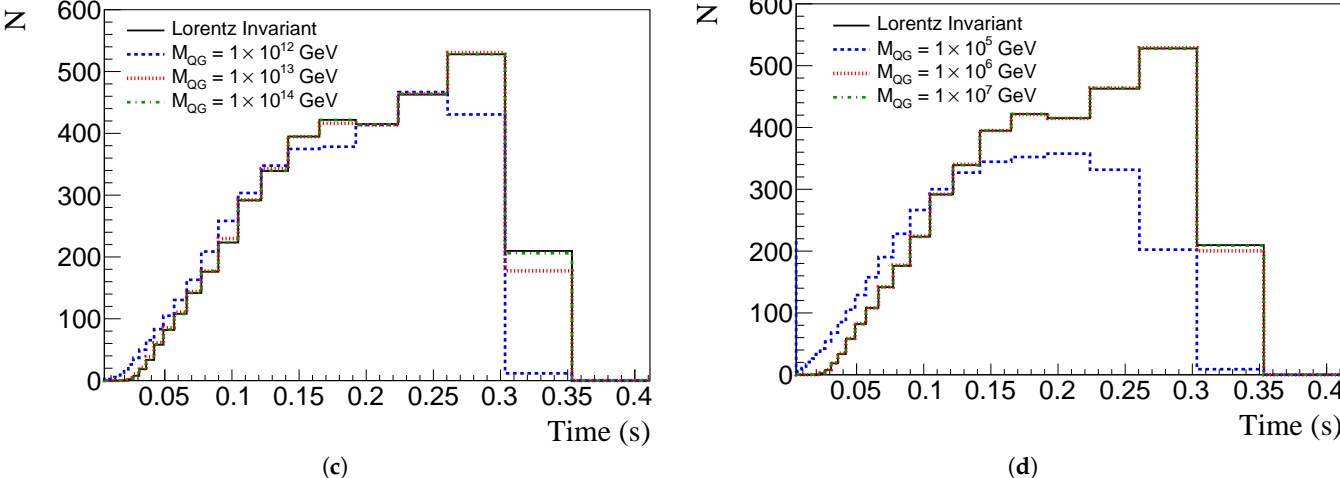

**Figure 7.** Event number per time bin in a 100 kton WtCh detector considering NO, different LIV energy dependence, $n$, and different energy scale, $M_{QG}$. Top (bottom) panels show subluminal (superluminal) effect. Left (Right) panels show $n = 1$ ($n = 2$). Blue dashed lines represent event numbers for $M_{QG} = 10^{12}$ GeV ($2 \times 10^5$ GeV), red dotted for $M_{QG} = 10^{13}$ GeV ($6 \times 10^5$ GeV), and green dash-dotted for $M_{QG} = 10^{14}$ GeV ($10^6$ GeV). Black solid line refers to conserved Lorentz invariance. (**a**) Subluminal; $n = 1$. (**b**) Subluminal; $n = 2$. (**c**) Superluminal; $n = 1$. (**d**) Superluminal; $n = 2$.

**Figure 8.** The same as in Figure 6 but for IO. (**a**) Subluminal; $n = 1$. (**b**) Subluminal; $n = 2$. (**c**) Superluminal; $n = 1$. (**d**) Superluminal; $n = 2$.

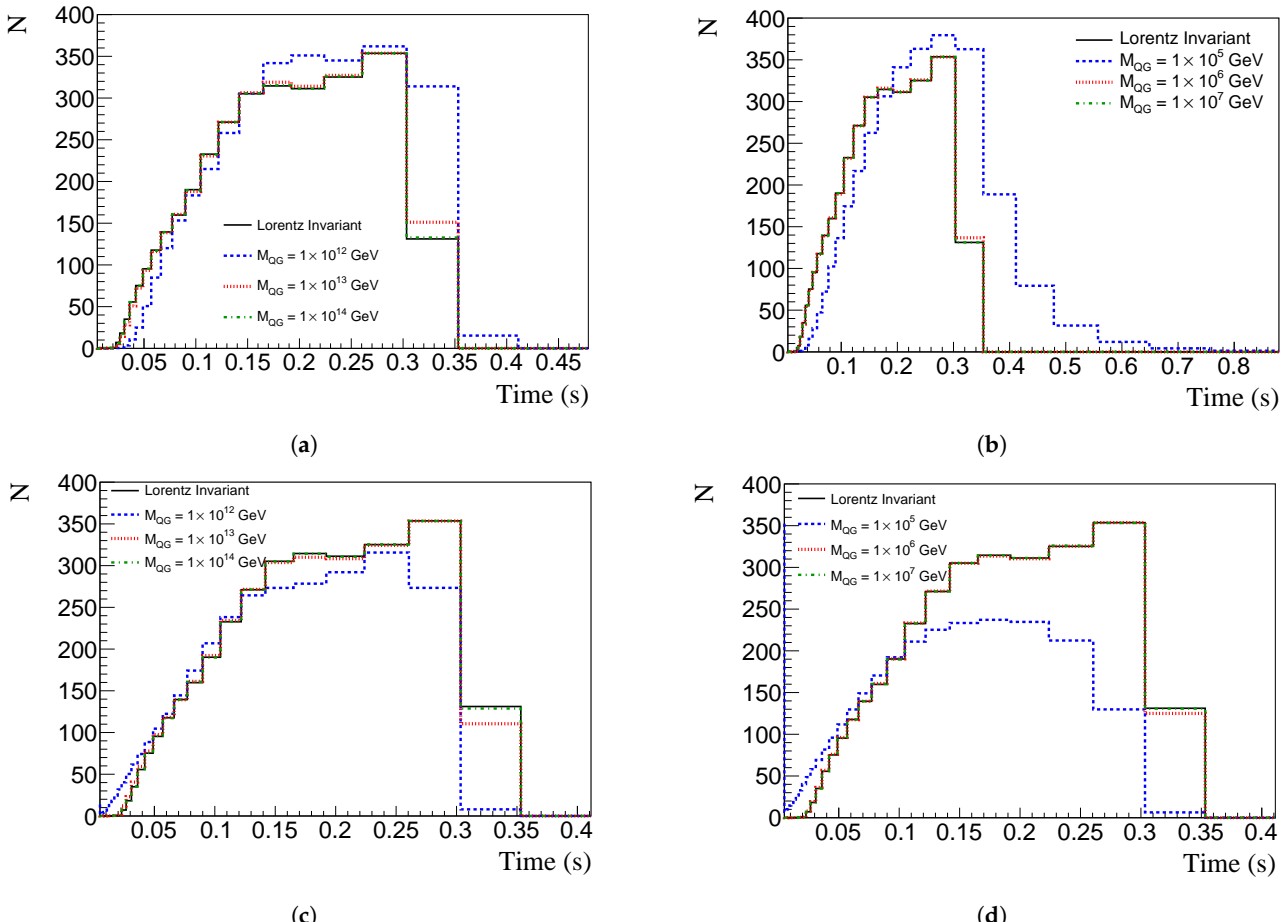

**Figure 9.** The same as in Figure 7 but for IO. (**a**) Subluminal; $n = 1$. (**b**) Subluminal; $n = 2$. (**c**) Superluminal; $n = 1$. (**d**) Superluminal; $n = 2$.

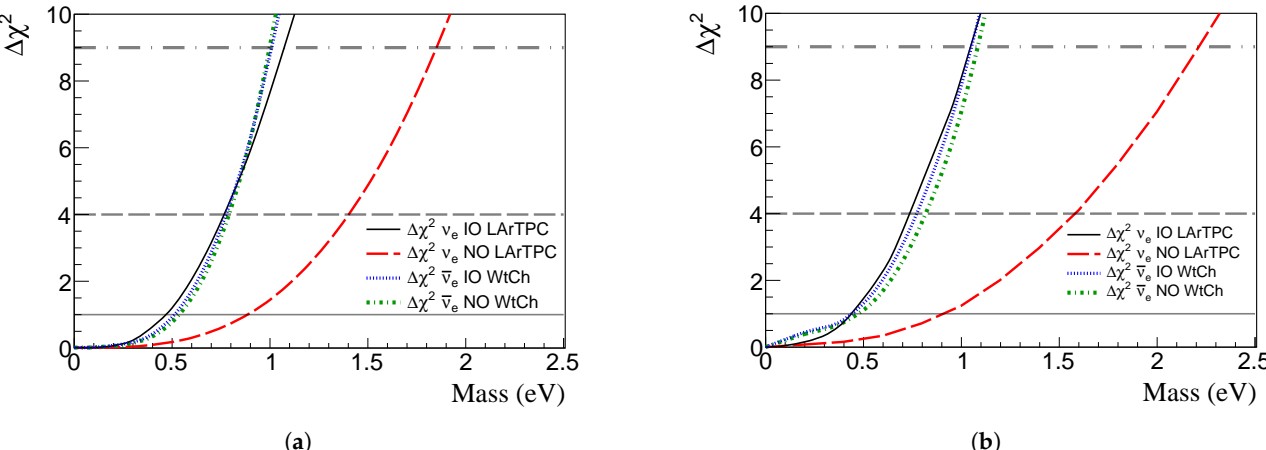

**Figure 10.** $\Delta\chi^2$ depending on $\nu$ mass considering $\nu_e$ ($\bar{\nu}_e$) event number in a 40 kton LArTPC (100 kton WtCh). Solid black (dotted blue) line and dashed red (dash-dotted green) line, respectively, are for IO and NO. SN at 10 kpc. (**a**) 15 $M_\odot$ SN. (**b**) 11.2 $M_\odot$ SN.

As expected, we see better limits for IO and $\nu_e$ in the DUNE-like detector because of the peak of neutronization. In the NO, the suppression of the neutronization peak worsens the sensitivity. For the HyperK-like detector, the limits on $\bar{\nu}_e$ for both mass orderings are very similar, as the time spectrum does not change significantly.

Our results for $\nu_e$ and $\bar{\nu}_e$ are summarized in Table 2. The numbers inside parentheses and without parentheses represent the mass limits for the LArTPC and WtCh detector, respectively.

**Table 2.** Limits on neutrino masses for a 100 kton WtCh (40 kton LArTPC) detector for NO and IO and 15 $M_\odot$ SN at 10 kpc. See Figure 10a.

| | | $\nu$ Mass (eV) | | |
|---|---|---|---|---|
| **Mass Ordering** | **$\nu$ Flavor** | **$1\sigma$** | **$2\sigma$** | **$3\sigma$** |
| NO | $\bar{\nu}_e$ | 0.52 (1.23) | 0.79 (1.93) | 1.00 (2.83) |
| IO | $\bar{\nu}_e$ | 0.49 (1.29) | 0.77 (2.11) | 1.01 (3.08) |
| NO | $\nu_e$ | 1.71 (0.88) | 1.90 (1.40) | 2.64 (1.85) |
| IO | $\nu_e$ | 0.68 (0.47) | 1.25 (0.77) | 2.01 (1.07) |

Some comments on the mass are important. Supposedly, the mass eigenstates of neutrinos travel freely through space and arrive incoherently at the detector; i.e., $\nu_1$, $\nu_2$, and $\nu_3$ arrive separately, and these mass eigenstates have, in fact, the mass information. Consider, for instance NO, where $\nu_1$ is the lightest mass eigenstate and reaches the detector first. If there is an interaction, the wave function collapses,[2] thus selecting the flavor state of the neutrino, such as, e.g., $\nu_e$, with a detection probability given by the PMNS matrix element $|U_{e1}|^2$.[3] If $\nu_1$ does not interact, the eigenstate $\nu_2$, which arrives at the detector with its respective delay associated with its mass $m_2$, may interact. The time interval between $\nu_1$ and $\nu_2$ can be estimated as [53]

$$\delta t = \frac{L}{2E^2}(m_2^2 - m_1^2). \tag{20}$$

Using $m_2^2 - m_1^2 \approx 7 \times 10^{-5}$ eV$^2$, $L = 10$ kpc, and $E = 10$ MeV in Equation (20), we obtain $\delta t \approx 10^{-7}$ s. For $m_3^2 - m_1^2 \approx 10^{-3}$ eV$^2$, $\delta t \approx 10^{-5}$ s. One needs a detector with time resolution better than $\delta t$ to allow the determination of the mass bound to each mass eigenstate. In our analysis, according to the expected experimental time resolution, we group the events in certain time intervals longer than the $\delta t$. So we put limits for an effective mass of the detected flavor.

The limits on LIV are shown in Figure 11a with the $\Delta\chi^2$ in terms of $M_{QG}$ for the 40 kton LArTPC. We show the superluminal and subluminal cases for NO and IO and $n = 1$ in Equation (18). The solid black curve represents the subluminal and IO case, the red dashed line represents the superluminal and IO case, the dotted blue curve is for the subluminal and NO, and the green dash-dotted curve is for the superluminal and NO.

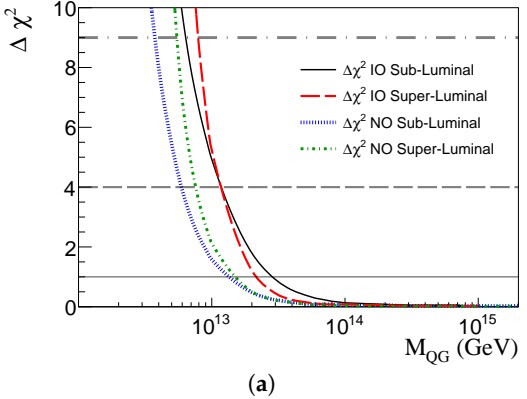
(a)

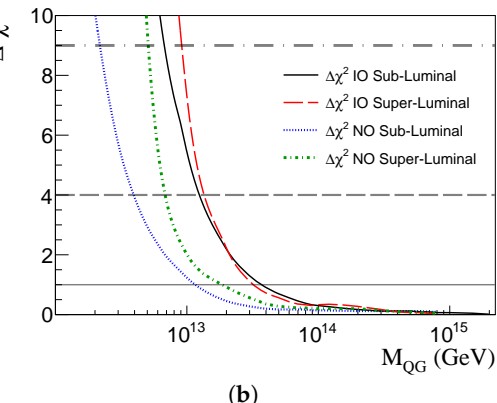
(b)

**Figure 11.** $\Delta\chi^2$ in terms of $M_{QG}$ for the superluminal and subluminal cases and NO and IO for $n = 1$ considering a 40 kton LArTPC. Black solid curve represents the subluminal and IO case, red dashed represents the superluminal and IO case, blue dotted curve is for the subluminal and NO, and green dash-dotted curve is for the superluminal and NO. SN at 10 kpc. (a) 15 $M_\odot$ SN. (b) 11.2 $M_\odot$ SN.

The $\Delta\chi^2$ has similar behavior compared to Figure 11a for $n = 2$ and for the WtCh detector for both values of $n$. Thus, we do not show those curves.

The inferior limits on the $M_{QG}$ for the subluminal and superluminal LIV effects are summarized in Tables 3 and 4 for $n = 1$ and $n = 2$, respectively. Numbers without (inside) parentheses are for WtCh (LArTPC) detectors.

**Table 3.** Inferior limits on LIV scale $M_{QG}$, $n = 1$ in Equation (18) for a 100 kton WtCh (40 kton LArTPC) detector, NO and IO, and the cases of subluminal and superluminal effects. An SN with 15 $M_\odot$ and 10 kpc from Earth is considered. See Figure 11a for the LArTPC case.

| | Mass Ordering | $M_{QG}$ ($\times 10^{13}$GeV) | | |
| | | $1\sigma$ | $2\sigma$ | $3\sigma$ |
|---|---|---|---|---|
| subluminal | NO | 2.9 (1.5) | 1.7 (0.6) | 0.9 (0.4) |
| subluminal | IO | 3.1 (2.9) | 1.7 (1.3) | 0.9 (0.7) |
| superluminal | NO | 3.1 (1.7) | 1.8 (0.8) | 1.1 (0.6) |
| superluminal | IO | 3.2 (2.3) | 1.8 (1.3) | 1.2 (0.8) |

**Table 4.** Inferior limits on LIV scale $M_{QG}$, $n = 2$ in Equation (18) for a 100 kton WtCh (40 kton LArTPC) detector, NO and IO, and the cases of subluminal and superluminal effects. An SN with 15 $M_\odot$ and 10 kpc from Earth is considered.

| | Mass Ordering | $M_{QG}$ ($\times 10^5$ GeV) | | |
| | | $1\sigma$ | $2\sigma$ | $3\sigma$ |
|---|---|---|---|---|
| subluminal | NO | 8.5 (5.9) | 5.9 (4.0) | 4.9 (3.3) |
| subluminal | IO | 8.6 (7.5) | 5.9 (4.8) | 4.8 (3.7) |
| superluminal | NO | 8.8 (6.2) | 6.3 (4.6) | 5.2 (3.9) |
| superluminal | IO | 8.8 (6.9) | 6.3 (5.0) | 5.2 (4.2) |

We can use the same methodology for other SN models, and we choose one model with 11.2 $M_\odot$ [42]. This model includes a longer period of neutrino emission of about 10 s, which includes the cooling time in the SN explosion. The main purpose here is to verify the impact of the later neutrino emission times on the bounds of the mass and LIV scale. The time spectrum and $\Delta\chi^2$ are very similar to the ones presented in the 15 $M_\odot$ SN case. In Figures 10b and 11b, we present for the 11.2 $M_\odot$ SN, the $\Delta\chi^2$ in terms of neutrino mass (LIV energy scale). Similar to the neutrino mass, the LIV $\Delta\chi^2$ for $n = 2$ and for the WtCh detector for both values of $n$ have similar behaviors compared to the one in Figure 11b. Thus, we do not show the $\Delta\chi^2$ plots for those cases but only provide tables with the results.

Our results using the 11.2 $M_\odot$ SN model for the $\nu_e$ and $\bar{\nu}_e$ mass limits are shown in Table 5. The numbers inside and without parentheses show limits for LArTPC and WtCh, respectively.

**Table 5.** Limits on neutrino masses for a 100 kton WtCh (40 kton LArTPC) detector for NO and IO and 11.2 $M_\odot$ SN at 10 kpc. See Figure 10b.

| Mass Ordering | $\nu$ Flavor | $\nu$ Mass (eV) | | |
| | | $1\sigma$ | $2\sigma$ | $3\sigma$ |
|---|---|---|---|---|
| NO | $\bar{\nu}_e$ | 0.45 (1.71) | 0.82 (3.13) | 1.08 (4.64) |
| IO | $\bar{\nu}_e$ | 0.43 (1.66) | 0.77 (3.27) | 1.05 (5.05) |
| NO | $\nu_e$ | 1.31 (0.92) | 2.43 (1.58) | 3.71 (2.22) |
| IO | $\nu_e$ | 0.65 (0.44) | 1.23 (0.74) | 2.43 (1.04) |

Following the same steps previously mentioned, the inferior limits on the $M_{QG}$ considering the subluminal and superluminal LIV effects and using the 11.2 $M_\odot$ SN model are summarized in Tables 6 and 7 for $n = 1$ and $n = 2$, respectively.

**Table 6.** Inferior limits on LIV parameter $M_{QG}$, $n = 1$ in Equation (18) for WtCh (LArTPC) detector considering NO and IO and the cases of subluminal and superluminal effects for 11.2 $M_\odot$ SN, 10 kpc from Earth. See Figure 11b for the LArTPC case.

| | Mass Ordering | $M_{QG}$ ($\times 10^{13}$ GeV) | | |
| | | $1\sigma$ | $2\sigma$ | $3\sigma$ |
|---|---|---|---|---|
| subluminal | NO | 4.1 (1.3) | 1.4 (0.4) | 0.6 (0.2) |
| subluminal | IO | 8.7 (3.8) | 2.0 (1.4) | 1.2 (0.7) |
| superluminal | NO | 11.0 (1.9) | 1.7 (0.7) | 0.9 (0.5) |
| superluminal | IO | 5.1 (3.2) | 1.8 (1.7) | 0.9 (0.9) |

**Table 7.** Inferior limits on LIV scale $M_{QG}$, $n = 2$ in Equation (18) for WtCh (LArTPC) detector considering NO and IO and the cases of subluminal and superluminal effects for 11.2 $M_\odot$ SN, 10 kpc from Earth.

| | Mass Ordering | $M_{QG}$ ($\times 10^5$ GeV) | | |
| | | $1\sigma$ | $2\sigma$ | $3\sigma$ |
|---|---|---|---|---|
| subluminal | NO | 8.6 (4.7) | 4.6 (2.9) | 3.4 (2.1) |
| subluminal | IO | 14.6 (7.7) | 6.1 (4.7) | 4.4 (3.2) |
| superluminal | NO | 9.9 (5.6) | 5.4 (4.0) | 4.5 (3.7) |
| superluminal | IO | 9.4 (7.9) | 5.5 (5.3) | 4.7 (4.4) |

The above results show that for obtaining the neutrino mass limit there is no significant impact when we compare both SN progenitor star masses with different neutrino emission time periods.[4] For $\nu_e$, the best limit comes from the DUNE-like detector and IO due to the neutronization burst. In fact, different progenitor masses do not significantly modify the neutronization phase [55]. This, somehow, demonstrates the relevance to study well the neutronization phase for determining neutrino properties, such as its mass. For the $\bar{\nu}_e$, the best limit comes from the HyperK-like detector, and neither mass ordering presents significant differences for their limits.

For the LIV scale sensitivity, we obtain very similar limits with both detectors and both mass orderings both for $n = 1$ or $n = 2$. This is related to the fact that LIV affects equally the different neutrino mass eigenstates and that the analysis is conducted assuming all detection channels. There is no significant difference in the results for the SN models that we consider. The inferior limits obtained by Chakraborty et al. [40] for a Mton water Cherenkov detector and using the neutronization burst signal are $M_{QG} \sim 10^{12}$ GeV ($M_{QG} \sim 2 \times 10^5$ GeV) for $n = 1$ ($n = 2$). They are very similar to ours—see Tables 3–7—however, they did not conduct a statistical analysis.

Before the conclusions, we would like to point out some important details to be explored in future analyses: First, there is the necessity of extending our analysis to other simulations of core-collapse SNs so we can explore the effective impacts of astrophysical parameters of the explosion and their relation to the bounds on neutrino masses and LIV. In addition, other and more massive progenitor stars can produce more events from accretion and cooling [56]. Thus, a deeper relation among astrophysical and neutrino parameters can be explored. Second, even though in Ref. [48] the authors pointed out that background or uncertainties on neutrino production, propagation, and interaction do not seem to significantly impact their parameter limits, it seems reasonable to explore deeply the uncertainties when more details are learned from the detectors developed for experiments such as DUNE and HyperK.

## 6. Conclusions

Summarizing our results, we conduct an analysis comparing the time spectra of supernova neutrino events, taking as a baseline the expected event rate for massless neutrinos and invariance under Lorentz transformation. We compute the event rate for two

kinds of detectors and two supernova progenitor star masses considering their location at 10 kpc from the detectors. We also analyze the two neutrino mass ordering possibilities.

Comparing the two supernova models, we did not see a big influence in the results coming from different masses or considering later neutrino emission times with events from the cooling phase of the supernova.

The two kinds of detectors, one a 40 kton liquid argon TPC similar to DUNE and the other a 100 kton water Cherenkov light detector similar to Hyper-Kamiokande, are sensitive to different neutrino interaction channels. The former is especially sensitive to electron neutrinos, which are produced in great amounts during the neutronization phase of the supernova, while the latter is sensitive to electron antineutrinos through inverse beta decay.

We observe a balance between the event rate during the neutronization phase, for which a DUNE-like detector is more sensitive if inverted mass ordering happens to be the case because the detector mass is more than two times bigger for a Hyper-Kamiokande-like detector. We verify this fact, obtaining the best limits on the neutrino effective masses of approximately 1 eV at $3\sigma$. The best antineutrino mass limit comes from a Hyper-Kamiokande-like detector and is essentially the same independent of the mass ordering, while the best neutrino mass limit comes from a DUNE-like detector for inverted mass ordering.

We also put limits on Lorentz invariance violation considering the energy scale at which Lorentz invariance could be broken. We sum the event rate over all neutrino flavors given that the same violating effect would influence their propagation. It is possible to distinguish between a superluminal and a subluminal violation effect, even though the constraints on the energy scale are similar. For linear energy dependence, the sensitivity is higher, and we find an energy scale limit of $M_{QG} \sim \mathcal{O}(10^{13})$ GeV. For quadratic energy dependence, we find a limit of $M_{QG} \sim \mathcal{O}(10^{5})$ GeV. These limits are compatible with results elsewhere, but here, we conduct a more-accurate statistical analysis.

**Author Contributions:** Conceptualization, C.A.M.; Methodology, C.A.M.; Software, L.Q.; Validation, L.Q.; Formal analysis, F.R.-T.; Writing—original draft, F.R.-T.; Writing—review & editing, C.A.M. and F.R.-T.; Supervision, C.A.M. All authors have read and agreed to the published version of the manuscript.

**Funding:** L. Quintino thanks the partial financial support provided by the Coordenação de Aperfeiçoamento de Pessoal de Nível Superior—Brasil (CAPES)—Finance Code 001.

**Institutional Review Board Statement:** Not applicable.

**Informed Consent Statement:** Not applicable.

**Data Availability Statement:** Not applicable.

**Acknowledgments:** F. Rossi-Torres thanks UFABC for the hospitality.

**Conflicts of Interest:** The authors declare no conflict of interest.

## Abbreviations

The following abbreviations are used in this manuscript:

| | |
|---|---|
| SN | Supernova |
| NO | Normal ordering |
| IO | Inverted ordering |
| LIV | Lorentz Invariant Violation |
| CC | Charged Current |
| NC | Neutral Current |
| ES | Elastic Scattering |
| IBD | Inverse Beta Decay |
| DUNE | Deep Underground Neutrino Experiment |

| HyperK | Hyper-Kamiokande |
|--------|------------------|
| SK | Super-Kamiokande |
| MSW | Mikheyev–Smirnov–Wolfenstein |
| LArTPC | Liquid Argon Time Projection Chamber |
| PMNS | Pontecorvo–Maki–Nakagawa–Sakata |
| WtCh | Water Cherenkov |

## Notes

1.  The experiment performed in this work has no specific name, so we refer in this table to the first name author.
2.  See Ref. [52] for a more precise discussion about the theory of neutrino detection.
3.  See Ref. [1] for a recent value of $|U_{e1}|^2$ and other neutrino oscillation parameters.
4.  Other analysis using WtCh detectors such as HyperK showed that it is possible to distinguish between SN simulation models with different neutrino emission mechanisms considering times of neutrino emission up to $\approx 9$ s [54]. They did not consider MSW on the neutrino propagation.

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
