# Peer review of "Analyzing the Time Spectrum of Supernova Neutrinos to Constrain Their Effective Mass or Lorentz Invariance Violation"

_universe, doi:10.3390/universe9060259_

Round 1

Reviewer 1 Report

The Authors have presented a detailed analysis of supernova neutrinos for constraining Lorentz Invariance Violation.

Author Response

Thank you for your report.

Reviewer 2 Report

This is a well-written paper that is well written. It provides guidelines and methods for calculating the neutrino-mass limits for future experiments.

Author Response

Thank you for your report.

Reviewer 3 Report

This manuscript presents an analysis of the expected arrival time spectrum of supernova neutrinos with two models, using two detection techniques- LArTPC and water Cherenkov detector. The best mass limits are set for both cases. It is overall well written. However, some clarifications are needed.

Line 30, From matter effect -> From the matter effect.

Line 49, ways for measuring -> ways to measure.

Line 88, Liquid Argon TPC(LArTCP) -> Liquid Argon Time Projection Chamber (LArTPC)

Line 92, please elaborate. What types of events are being described here?

Line 98, please add a reference for Hyper-K.

Equation 1, please define E.

Lines 109-115, section -> Section.

Line 123 and Figure 1, please define \nu_x.

Line 124, model dependent -> model-dependent.

Line 150, both dependent -> both of which depend.

Line 182, LArTPC has been defined earlier. Please add a reference.

Figure 10, DUNE and HK in the legend should be referred to as LArTPC and WtCh, just to be consistent with the text.

Line 314, please elaborate on "uncertainties". 

Line 331, later -> latter.

References

3, there are two papers under it, please fix it.

41, can not open the link, please fix it.

Some suggestions are included in the above section.

Author Response

Dear reviewer, please find below the list of corrections and changes according to your comments:

a-) Line 30, From matter effect -> From the matter effect.

Corrected.

b-) Line 49, ways for measuring -> ways to measure.

Corrected.

c-) Line 88, Liquid Argon TPC(LArTCP) -> Liquid Argon Time Projection Chamber (LArTPC)

Corrected.

d-) Line 92, please elaborate. What types of events are being described here?

events -> electron neutrino events with energies around 10 MeV through charged current (CC) interaction

Line 188, charged current (CC) -> CC, because now it is in the defined above.

e-) Line 98, please add a reference for Hyper-K.

We added reference [33], which appeared further in the same paragraph and is related to the Hyper-K design report.

f-) Equation 1, please define E.

Included the definition just after the mean neutrino energy.

g-) Lines 109-115, section -> Section.

Corrected.

h-) Line 123 and Figure 1, please define \nu_x.

Defined nu_x in line 123 and in the figure caption. We also corrected one misspelling:

the two models. -> for the two models. $\nu_x$ is defined as the sum of all muon and tau neutrinos and antineutrinos, i.e., $\nu_\mu + \bar\nu_\mu + \nu_\tau + \bar\nu_\tau$.

i-) Line 124, model dependent -> model-dependent.

Corrected.

j-) Line 150, both dependent -> both of which depend.

Corrected.

k-) Line 182, LArTPC has been defined earlier. Please add a reference.

Corrected. We added the following reference at the end of the sentence:

  1. Rubbia, “The Liquid Argon Time Projection Chamber: A New Concept for Neutrino Detectors,” Tech. Rep. CERN-EP-INT-77-08, 1977

l-) Figure 10, DUNE and HK in the legend should be referred to as LArTPC and WtCh, just to be consistent with the text.

Legend corrected.

m-) Line 314, please elaborate on "uncertainties". 

We elaborate as follows:

several uncertainties -> background or uncertainties on the neutrino production, propagation, and interaction

n-) Line 331, later -> latter.

Corrected.

o-) References

3, there are two papers under it, please fix it.

Fixed.

41, can not open the link, please fix it.

We corrected the link.

We also corrected in the last sentence of the Conclusions: This limits -> These limits

Thank you for your report. We really appreciate it. For the PDF file with the changes in the article, please see the attachment.

Reviewer 4 Report

This paper is very well written and presents interesting and original results. My suggestion is to publish it as it is.

Author Response

Thank you for your report.

Reviewer 5 Report

The presented study is of significant value for the future experiments such as DUNE and HyperK. Therefore, I believe that it is important to clarify a few things in a way that would make this article even better.

My main concern is lack of explanation on the LIV effect. As it is one of the crucial alternative hypotheses to probe, I find it disturbing that it is not touched in the Introduction section. 

The second concern is as following - from the review given in the Introduction it is quite clear that the neutrino mass effect has to be included in all feasibility studies. Therefore, I can not understand why authors study LIV influence separately from the mass effect. From Figure 2 it is tempting to say that two effects under certain circumstances can even cancel each other.

Minor comments and questions:

1) formula 1 - I would suggest to specify somewhere what index "0" means 

2) how is mean value <E> defined? 

3) figure 1 - how can non-zero luminosity prior to t=0 can be explained?

4) formula 11 and 12 - I would suggest to specify here that delta t depends explicitly on E

line 100. For HyperK, it is expected an absolute mass -> For HyperK, it is expected to have an absolute mass

line 123. luminosity time evolution the two models -> luminosity time evolution for the two models

figure 2 caption. presentend -> presented 

line 191. do not -> does not

Author Response

Dear reviewer,

Thank you for your comments. They really help us to improve our work. Please find below the tentative answers to your concerns and the corrections that were based on you suggestions and questions:

“My main concern is lack of explanation on the LIV effect. As it is one of the crucial alternative hypotheses to probe, I find it disturbing that it is not touched in the Introduction section.”

In order to give more information about LIV in the introduction, we brought part of the text in Sec.3.2 around Eq.(18) to the paragraphs just before the article organization paragraph. From line 110 to 118 we give more basic explanations with references. We added reference Antonelli et al. Symmetry 12 (2020) no.11, 1821.

“The second concern is as following - from the review given in the Introduction it is quite clear that the neutrino mass effect has to be included in all feasibility studies. Therefore, I can not understand why authors study LIV influence separately from the mass effect. From Figure 2 it is tempting to say that two effects under certain circumstances can even cancel each other.”

In our study, we decided to separate the two effects, of mass and LIV, because the electron (anti)neutrino effective mass may still be relatively small compared to the upper limits that we find. We can see from figure 10 that we have poor sensitivity for masses below 0.5 eV, which means that the effect of neutrino mass may be small enough so that we study the LIV effect without considering the neutrino mass. Now we explain this in lines 119-123. A study with both, mass and the LIV parameter, considered at the same time would allow the obtention of a sensitivity region in the 2D parameter space “mass vs. M_QG”. Even though it is very interesting and gives information on the interplay of the two effects, it would bring subtleties and complications to the analysis that are not within the scope of this article. This would deserve a whole new analysis.

1) formula 1 - I would suggest to specify somewhere what index "0" means 

We specify now just after the formula. It corresponds to the original flux.

2) how is mean value <E> defined? 

<E> is defined according to the neutrino energy spectrum, taking its average at each instant of time. After formula 1 we define E as the instant energy and <E> as the mean neutrino energy.

3) figure 1 - how can non-zero luminosity prior to t=0 can be explained?

In this figure we are showing the luminosity change as a function of time, but it is independent of where t=0 is. We can always redefine where the time t=0 is located.

4) formula 11 and 12 - I would suggest to specify here that delta t depends explicitly on E

We included the energy dependence on Delta t.

5) line 100. For HyperK, it is expected an absolute mass -> For HyperK, it is expected to have an absolute mass

Corrected

6) line 123. luminosity time evolution the two models -> luminosity time evolution for the two models

Corrected

7) figure 2 caption. presentend -> presented 

Corrected

8) line 191. do not -> does not

Corrected.

All the changes related to your report are highlighted in violet when we add words and strikethrough when we delete them. Other changes are in blue. Please find the revised version in the attachment. Thank you!

Round 2

Reviewer 5 Report

I would like to thank authors for the clarifications given in their response.

I believe that the introduced modifications improve the paper and it can be published in the present form.